# Influence of Implant Design and Under-Preparation of the Implant Site on Implant Primary Stability. An In Vitro Study

**DOI:** 10.3390/ijerph17124436

**Published:** 2020-06-20

**Authors:** Mariano Herrero-Climent, Bernardo Ferreira Lemos, Federico Herrero-Climent, Carlos Falcao, Helder Oliveira, Manuela Herrera, Francisco Javier Gil, Blanca Ríos-Carrasco, José-Vicente Ríos-Santos

**Affiliations:** 1Porto Dental Institute, 4150-518 Porto, Portugal; dr.herrero@herrerocliment.com (M.H.-C.); blemos@ufp.edu.pt (B.F.L.); cfalcao@ufp.edu.pt (C.F.); 2Faculty of Health Sciences, Fernando Pessoa University, 4249-004 Porto, Portugal; helderol@ufp.edu.pt; 3Department of Periodontology, University Complutense of Madrid, 28040 Madrid, Spain; clinica@fherrerocliment.net; 4Department of Stomatology, University of Seville, 41009 Seville, Spain; manuelah@us.es; 5Faculty of Dentistry, International University of Catalonia, 08017 Barcelona, Spain; xavier.gil@uic.cat; 6Department of Periodontology, University of Seville, 41009 Seville, Spain; brios@us.es

**Keywords:** dental implant, tapered, under-preparation, implant preparation, implant stability, insertion torque, ISQ, RFA

## Abstract

The aim of this study was to evaluate the effects of different implant sites an under-preparation sequence associated with two different implant designs on implant primary stability measured by two parameters: insertion torque (IT) and implant stability quotient (ISQ). It used two different implants: one cylindrical as a control and another one with a tapered design. The implants were inserted in type III fresh humid bovine bone and four drilling sequences were used: one control, the one proposed by the implant company (P1), and three different undersized (P2, P3 and P4). P2 was the same as P1 without the cortical drill, P3 was without the last pilot drill and P4 was without both of them. The sample size was *n* = 40 for each of the eight groups. Final IT was measured with a torquemeter and the ISQ was measured with Penguin resonance frequency analysis. Results showed that both ISQ and IT have a tendency to increase as the preparation technique reduces the implant site diameter when compared with the standard preparation, P1. The preparations without cortical drill, P2 and P4, showed the best results when compared with the ones with a cortical drill. Tapered implants always showed higher or the same ISQ and IT values when compared with the cylindrical implants. Giving the limitations of this study, it can be concluded that reducing implant preparation can increase IT and ISQ values. Removing the cortical drill and the use of a tapered design implant are also effective methods of increasing primary implant stability.

## 1. Introduction

Implant primary stability [IPS) is a mechanical concept that has been defined as the degree of tightness of a dental implant immediately after placement in its prepared osteotomy; an implant is considered to present initial stability when it is clinically immobile at the moment of placement [1] IPS is widely accepted as a prerequisite to achieve osseointegration [2,3]. Despite so, at the early days of osseointegrated implantology, the conventional protocols followed a two-surgeries approach in such a way that the implants were submerged under the mucosa looking for an undisturbed bone healing [4,5]. However, the development of different treatment modalities, like a) the use of per-mucosal healing implants, i.e., implants with polished neck [6,7], b) the modification of the conventional protocol by using original two-stage implants in one single stage by means of the placement of abutments or high healing caps after the implant insertion [8,9], or c) the placement of a prosthesis just immediately or in the following days after the implant insertion with or without occlusal contact [10,11], made of IPS is a critical aspect to be considered.

Although implant therapy has shown excellent survival rates over the years, there are still some biological and biomechanical problems associated with this procedure, such as implant failure, implant overload, screw loosening, etc. [12,13].

Many factors have been proposed and are already shown in the literature to be involved in the IPS [14,15,16]. Some of them are inherent to each particular situation, like the bone quality [17], the implant length [18], diameter [19], or the experience of the operator [20]. However, there are others, like the shape of the implant [21], the presence of threads [22] or geometry [23], and the implant bed preparation technique [24] that are not case-dependent, and the decisions on them may influence the IPS. Other aspects like implant surface roughness seems to have no effect on IPS [25]. 

An intentional under-preparation of the implant osteotomy, in order to promote a more frictional insertion of the implant, is not recent, and it has been already documented in the literature [26,27]. However, and considering the many factors involved and the singularity of each of the implant sites, any standardization or fixed protocol is hard to be considered, and it turns into a matter of clinical decision. In addition, at some degree the uncontrolled increase of IPS would represent a detrimental factor for the marginal bone level preservation due to excessive bone compression [28,29]. Ideally, the operator should try to achieve a high value of IPS by avoiding marginal bone stressing.

The implant industry has shown an increased interest in the development of new implant designs, mostly addressed to increase the IPS for the so-called immediate loading treatments [30,31]. Mainly tapered implants or its modifications have been shown to provide higher IPS when compared to cylindrical or non-tapered designs [21,32,33]. When considering immediate loading in which IPS is a determinant factor, there seems to be a general agreement that tapered or conical implants will provide better results [10]. Finally, the implant thread design, as already said, also plays a role in the final IPS, and this could be of clinical relevance for immediate loading [23].

In addition, it could be of interest for the clinician to know the impact of the changes on the implant designs owing to the wide offer of commercially available tapered implant design, and, furthermore, the impact that would have on the modification of the drilling sequence on the IPS.

In today’s clinical practice, it is usual to use implants with a bone-level concept and switch platform treatment philosophy in order to achieve a better preservation of the best crestal and to achieve a better aesthetic of the final rehabilitating result. With the use of immediate loading protocols, and in cases of post-extraction with lower bone density situations, the clinician seeks a greater primary stability of the implants, and it can be achieved by using an implant with a design that provides higher stability or altering of the milling sequence [34], or even a combination of both. Attempting to help answer these questions would be the aim of the present study.

The aim of this study was to compare the behavior of two implant designs (cylindrical and tapered) with the same switching platform concept when different techniques for preparing the surgical site are used, evaluating the influence on the primary stability of the implant. Insertion torque (IT) values and resonance frequency analysis (RFA) values were recorded.

## 2. Materials and methods

### 2.1. Implants

Two different shapes of implants from the Klockner Implant System were used: one straight cylindrical implant as a control group, VEGA, and a new tapered implant with bone condensing capacity named VEGA X as test group (Figure 1). Both are bone-level-type implants, with the aim to use the switching platform philosophy of treatment. A transepithelial abutment is connected to the implant by an internal Morse cone connection, so the implants are a two-piece implant system.

Two implant diameters were used, 3.5 mm and 4.0 mm, both 10 mm in length. 

Both of the cylindrical and tapered implants used in this research had the same surface treatment, obtained by shot blasting and acid passivate. The measurements were made in three different surfaces to characterize the Ra (the average roughness), which is the arithmetic average of the absolute values of the distance of all points of the profile to the mean line (surface roughness, Ra: 1.30 ± 0,23 µm). The shot blasting procedure main objective is to produce a rough surface by means of blasting alumina particles with 185 µm diameter. The acid passivate has the objective of surface decontamination of the alumina remains with the goal of generating a thicker layer of titanium oxide in the surface. Roughness was evaluated for the test surfaces in the framework of the recommendations by Wennerberg and Albrektsson on topographic evaluation for dental implants [35]. 

### 2.2. Bone

This study was performed on fresh bovine kneecaps. Bovine kneecaps have a thinner portion of cortical bone and a larger proportion of medullar bone. According to the classification of Lekholm and Zarb, this is classified as bone type III density [36]. To avoid differences of bone density when comparing the two implants and the different types of site preparation, all the bone preparations were placed adjacent to each other. 

### 2.3. Sample Size

The sample size used was n = 40 in each study group. It was calculated with N Query Advisor v4.0 (Statsols, California, Los Angeles; EEUU) for p < 0.05 based on the research of Herrero-Climent. The calculated sample size was *n* = 18; the authors decided to increase the sample size (*n* = 40) to look for greater statistical relevance [37,38]. 

### 2.4. Ethical Committee

As this was an in vitro study made with bone of fresh bovine origin, the request of an ethical committee was not necessary.

### 2.5. Implant Site Preparation

Four different types of implant site preparations were performed with different types of under-preparation. The manufacturers proposed a preparation sequence and this was used for both types of implants as a reference or control.

Both implant types were placed in each type of preparation in such way that the implant surface to be in contact with the bone site was different depending on the type of preparation carried out. Both the type of implant site preparation and the design of the implant used or the combination of the two factors may have an impact on the primary stability of the implant placed.

The preparations in the bone fragments were all performed by the same surgeon with extensive experience in the use of the Klockner Implant System (more than five years) (Figure 2). They were performed by trying to simulate the application of the surgical techniques to be evaluated in daily clinical practice. All preparations were performed in a room with a stable temperature (22 °C). Acclimation of the bovine bone fragments was allowed for one hour before the preparations were made. The bone fragments were stored in a refrigerator. The site’s drilling was performed by trying to simulate the usual clinical situation by using an electronic surgical unit (W&H Dentalwerk, Austria), with abundant irrigation of physiological serum stored at a temperature of 6 °C. 

In order to standardize and control the depth of the implant beds, depth stops were used in all full-length drills; cortical drills do not reach the entire preparation and are not active in the most apical portion, leading to the fact that it was not necessary to use the drill stop.

A total of 640 implant site preparations were performed with the consequent insertion of 640 implants, distributed according to the groups described below.

### 2.6. Study Groups

I. According to the surgical technique or the bone preparation (40 implant site preparations for each implant type and for each implant diameter):P1: standard preparation and the recommended one by the company—control group.P2: the same as P1 but does not include the cortical drill (ref nº 18 02 04—3.5 mm ø implants/ref. 18 02 05—4.0 mm ø implants).P3: horizontal under-preparation technique that does not include the last full-length drill (ref nº 10 02 05 for 3.5 mm ø and ref. nº 10 02 06 for 4.0 mm diameter implants) but includes the cortical one.P4: horizontal under-preparation technique like P3 but without cortical drill (ref nº 18 02 04—3.5 mm ø implants/ref 18 02 05—4.0 mm ø implants).

II. According to the type of implant (Figure 3.):A. Cylindrical—VEGA implant○3.5 mm implant diameter○4.0 mm implant diameterB. Tapered - VEGA X implant○3.5 mm implant diameter ○4.0 mm implant diameter

### 2.7. Implant Stability Measurements

The primary stability obtained from the implants was assessed according to two variables: the resonance frequency analysis (RFA), and the insertion torque (IT). 

Once the bone sites were prepared, the implants were inserted with an analogic torquemeter Tohnichi—ATG6C (Tohnichi Mfg. Co Ltd., Japan) that recorded the final value of insertion torque.

The measurement of resonance frequency analysis was done with the Penguin RFA system (Integration Diagnostics, Sweden). RFA is described in the literature as a method for evaluating implants. The RFA values were registered by obtaining two perpendicular measurements, with the Penguin probe parallel to the bone surface. To obtain the ISQ (implant stability quotient), values of RFA and a transducer (MultiPeg) must be placed in the implant that allows the stability of the implant to be read. This device is specific for each type of implant and supplied by the manufacturer. The technique for placing the Multipeg and obtaining the RFA record was that suggested by the manufacturer. Subsequently, after the mean value was obtained after the two RFA records, the valid ISQ value was considered. For both implant designs for the 3.5 mm diameter implant, the MultiPeg used was number 57—ref. 55065, and for both implant designs for the 4.0 mm diameter implant, the MultiPeg number was 26—ref. 55034. All measurements were recorded by the same clinician who was unaware of the composition of the study groups and were taken immediately after the implant placement.

### 2.8. Drilling Sequences

The preparation sequence for 3.5- and 4.0-mm diameter implants proposed by the manufacturer is:Lanceolate drill/decortication for the first 6 mm (ref. nº 10 02 01)2.35 mm diameter (ø) initiation drill (ref. nº 10 02 02)2.8 mm ø pilot drill (ref. nº 10 02 03)3.5 mm ø cortical drill (ref. nº 18 02 04)3.3 mm ø pilot drill (ref. nº 10 02 05)—last of 3.5 mm diameter implant4.0 mm ø cortical drill (ref. nº 18 02 05)3.6 mm ø pilot drill (ref. nº 10 02 06)—last of 4.0 mm diameter implant

The last drill of each drilling sequence to be evaluated in the present study is shown in Figure 4 for 3.5 mm and 4.0 mm diameters for both implant designs. Preparation for 4.0 mm diameter implants requires the use of a 3.6 mm diameter pilot drill and a 3.95 mm cortical drill; after the use of the 3.3 mm diameter pilot drill, the rest of the sequence is the same for both implant types.

All implant site preparations in this study were made with the aim that the implants remained 1 mm subcortical once inserted into the bone.

### 2.9. Statistical Analysis

Statistical analysis was performed with Minitab 16 Statistical Software (Minitab release 13.0; Minitab Inc., Coventry, UK), using a 5% significance level. A normality test was conducted in all the samples studied in order to decide whether or not to use parametric or non-parametric statistics. 

Given the sample size (*n* = 40), it was considered that data not meeting the normal distribution were to be analyzed following non-parametric statistics. Accordingly, the next situations were found:For data meeting a normal distribution, two independent data groups were compared using a parametric t-student test, while three or more data groups were compared using ANOVA.For data not distributed normally, a Mann–Whitney test was used to compare two data groups and the Kruskall–Wallis test was used to compare three or more data groups. Non-parametric tests were also used to compare data groups meeting a normal distribution with data groups not meeting a normal distribution.

## 3. Results

Every implant reached its final position without any particular problem during the insertion procedure. Table 1 presents mean, standard deviation and *p* values of IT and ISQ for both 3.5 and 4.0 mm, as well as cylindrical and tapered implant designs recorded in the four types of site preparation techniques.

As it is shown in Table 1, almost all values of IT and ISQ are higher in the tapered one when comparing with the cylindrical one in every type of preparation technique, except ISQ for P3 and P4 on 4.0 mm diameter implants. 

Table 2 shows the statistical analysis from the first part of the study which was the influence of implant design on stability. In P1, both IT and ISQ in 3.5 mm and 4.0 mm diameter implants were higher in tapered design, *p* <0.005. In P2 in both 3.5 mm and 4.0 mm, only IT was statistically significantly higher, *p* < 0.05. In P3 the 3.5 mm implants tapered design showed higher values on both IT and ISQ, and for 4.0 mm only higher values were shown on IT, *p* < 0.05. The P4 only revealed higher values for 4.0 mm IT for the tapered design, *p* < 0.05. Statistical analysis showed that cylindrical implant was not higher for any of the different preparation techniques.

By comparing the preparation techniques and removing the cortical drill in P2 and P4 to compare with the groups with cortical drill (P1 and P3), it was seen that there was a significant improvement of the ISQ and IT values, especially when comparing P2 with P1 in both 3.5 mm (P1/Cylindrical IT: 34.6 Ncm; ISQ: 74.9; P2/Cylindrical IT: 44.1 Ncm; ISQ:78.2; P1/Tapered IT: 54.2 Ncm; ISQ: 78.2; P2/Tapered IT: 57.3 Ncm; ISQ: 78.6) and 4.0 mm implants (P1/Cylindrical IT: 43.6 Ncm; ISQ: 76; P2/Cylindrical IT: 45.2 Ncm; ISQ:79.1; P1/Tapered IT: 64.7 Ncm; ISQ: 78.5; P2/Tapered IT: 65.7 Ncm; ISQ: 79.9).

When comparing the ISQ values (A and B) in the same preparation technique, there was not any statistical difference between them.

## 4. Discussion

The increased use of immediate loading protocols led to the awareness and necessity of using methods that improve primary stability of implants, particularly in unfavorable situations like low-density bone sites or post-extractional implants. Over the years some protocol modifications have been proposed to overcome these difficulties. Under-preparation of the implant site seems to be one of the most efficient and easiest ways to increase the primary stability [15,39]. Other ways of improving implant stability, such as modifying the implant design from conical to tapered designs, have also been proven to be efficient in increasing implant stability [31,40,41]. This paper tried to relate both implant design and under-preparation of implant sites and evaluate the results on insertion torque and implant stability quotient values.

The present study attempted to evaluate the influence of a tapered implant and an actual trend and compare it with a cylindrical implant with the same treatment philosophy: bone-level type and switching platform concept. In recent years, similar studies have been published in which implant stability is assessed with different implant designs or even with different implant concepts, such as bone level and tissue level. The purpose of this study was to evaluate the design modifications, mainly on the implant core, combining with under-preparation of the implant bed and its influence on the primary stability [42,43,44].

In the values presented in Table 1, it can be clearly seen that there was an increase of IT and ISQ values when there was a reduction of the implant site, from normal preparation to the horizontal undersizing of both the pilot and the cortical drill. When comparing both designs in Table 1 it is also clear that the tapered design with the same preparation technique most of the time had higher values of IT and ISQ, essentially in the cases that use the cortical drill as a step of the site preparation (P1 and P3), showing that when we use a tapered implant it may not be necessary or mandatory to use the undersize osteotomy. 

An in vitro study published by Moon, 2010 had the same purpose of this one, to evaluate the effects of implant shape (straight and tapered type) and bone preparation on the primary stability. They prepared two types of bone (II and IV, both on bovine ribs) with three different protocols for each implant: under-preparation (−1 mm of height prep.), standard preparation and over-preparation (+1 mm). Compared to this study, the principal difference in the methodology was that the under-preparation in Moon’s study was vertical and this study evaluated horizontal under-preparation of the total diameter, and both led to an increase of the IPS because of the increase surface area of the implant in contact with the bone. They evaluated ISQ after each implant placement. The results found in this study are quite similar to the those found in Moon’s study in that the under-preparation group with the tapered design implant had higher ISQ values, with *p* < 0.05, when comparing to the straight design on both type II (76.83 ± 4.89/72.23 ± 8.72) and type IV bone (66.37 ± 6.54/61.40 ± 7.09) [45].

In 2017, Degidi, in an in vitro study, also evaluated a modification of the implant site preparation, stepped osteotomy (undersize only of the apical area to avoid excessive compression of the crestal bone), and compared it with a standard preparation and tapered design implant. The variables that were evaluated were the variable torque work (VTW), peak IT (pIT) and RFA (ISQ values). When comparing VTW and pIT it was concluded that the stepped osteotomy (VTW = 2280.53 ± 548.65 Ncm/pIT = 31.97 ± 8.98 Ncm) compared with standard osteotomy (VTW = 1919.29 ± 579.65 Ncm/pIT = 24.67 ± 8.99 Ncm) and tapered implants (VTW = 1620.62 ± 458.96/pIT = 22.5 ± 7.53) was the protocol with higher values of both variables *p* < 0.05. When comparing the ISQ values, stepped osteotomy (78.17 ± 4.22) revealed higher values when comparing with tapered design implants [75.98 ± 4.49), *p* < 0.05. However, this study didn’t compare the undersizing of stepped osteotomy in straight vs. tapered design. The results were quite similar to the ones found in the present study; the increase of pIT on the stepped osteotomy comparing to the standard osteotomy (pIT = 31.97 ± 8.98 Ncm/pIT = 24.67 ± 8.99 Ncm) was also found when comparing the IT of under-preparation with cortical drill (stepped osteotomy) on both designs on 3.5 mm diameter implants with the standard preparation (P3 vs. P1) (P1 Cylindrical: 34.6 Ncm/P3 Cylindrical: 63.7 Ncm/P1 Tapered: 54.2 Ncm/P3 Tapered: 75.9 Ncm) [46].

Another study published in 2010 by Bilhan’s group shows similar results to those found in this study. The methodology of the study was similar to this one. The aim of Bilhan’s study was to evaluate the effects of some surgical and implant-related factors in the improvement of primary stability and possibly the correlation with both RFA and IT. Like the results of this paper, they found that both cylindrical and tapered implants had higher ISQ and IT values with under-preparation drilling when comparing with the standard drilling (IT and ISQ cylindrical, standard drilling: 15.37 Ncm, 75.37 ISQ/underdimensioned drilling: 41.87 Ncm, 80.50 ISQ), (IT and ISQ tapered, standard drilling: 11.62 Ncm, 65.12 ISQ/underdimensioned drilling: 21.50 Ncm, 67.87 ISQ). The authors concluded that the under-preparation drilling technique improves the primary stability when the implants are inserted in low density bone type, type III and IV, according to Lekholm and Zarb’s classification [14,36]. 

A study in cadavers in 2015 from Boustany compared the differences between IT and RFA produced by conventional versus modified stepped osteotomies. The results showed that the modified stepped osteotomy had a significantly higher mean IT than the conventional protocol [26.8 Ncm and 15.9 Ncm, respectively, *p* <0.05), however no significant differences were found in RFA when comparing the two protocols [47]. These results are consistent with the results found in the present study where in some preparations there was not a correlation between IT and ISQ values; Table 2 shows 3.5 and 4.0 mm implant P2 preparation had a statistically significantly higher IT value, but not higher values of ISQ. These absence of correlation between IT and RFA has already been reported by several papers [18,48,49,50,51] and was confirmed by the results of this study [52,53].

The increase of IPS is found in under-preparation protocols because of the increased implant surface in contact with the bone, leading to higher ISQ and IT values as can be seen in both Table 1 and Table 2. The under-preparation aids bone condensing in the moment of the implant insertion in the implant bed so there is an increase in the amount of implant surface in contact with the bone.

The fact of not using a cortical drill could be an easy and predictable way of getting higher primary stability values, although it has been demonstrated that the pressure on the cortical area of the bone could be dangerous and may lead to bone loss around the implant neck. It is important for the clinician to evaluate the bone density and determine if there is necessity to remove the cortical drill to get even higher stability values, if it isn’t the stepped osteotomy shown in the P3 preparation could be a safer possibility for getting higher values of primary stability.

This study only evaluated mechanical aspects of the IPS. In the clinical environment there are several biological factors that can influence the IPS and its evolution to biological stability.

## 5. Conclusions

Within the limitations of an in vitro study, the following conclusions can be drawn:Under-preparation of the implant site can be a viable method to improve the implant primary stability on both ISQ and IT, since there is an increase of implant surface in contact with bone.The use of a tapered shaped implant can improve the implant stability when comparing to a cylindrical shaped implant, leading to the fact that it is not necessary to use of an under-preparation of the site combining the use of tapered implants on low density bone (type III/IV).The removing of the cortical drill from the standard preparation was the implant site protocol that showed the biggest improvement of the implant stability.At the same time the clinician must evaluate the density of the present bone and carefully judge the type of protocol for each case.

### Clinical Implications

As previously commented, the techniques that are being used in current implantology led to need for the clinician to have greater primary stability when placing implants for the success of the treatments, even more in cases of immediate loading protocols, post-extraction implants or implants placed in low-density bone. Clinically, the use of an implant design or a surgical technique that helps achieve greater stability may be sufficient, or a combination of both. According to the obtained results, under-drilling at the level of the cortical bone combined with the use of conical implants seems to be the recommended attitude when seeking higher primary stability.

## Figures and Tables

**Figure 1 ijerph-17-04436-f001:**
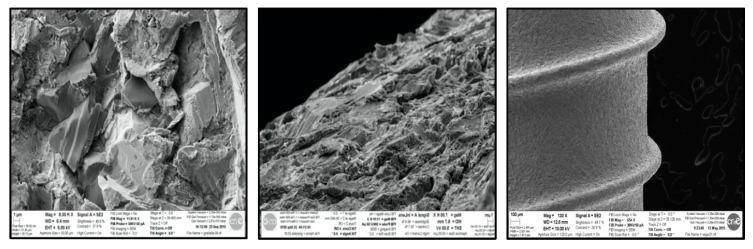
Implant surface, SEM photos of the shotblasted surface. It can be seen at different magnifications: first and second (1 µm) third (100 µm).

**Figure 2 ijerph-17-04436-f002:**
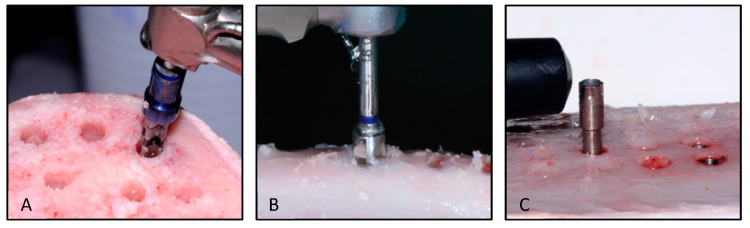
Implant site preparation;(**A**) —Pilot drill and 10 mm stop for implant bed depth control, (**B**)—cortical drill, (**C**)—RFA measurement.

**Figure 3 ijerph-17-04436-f003:**
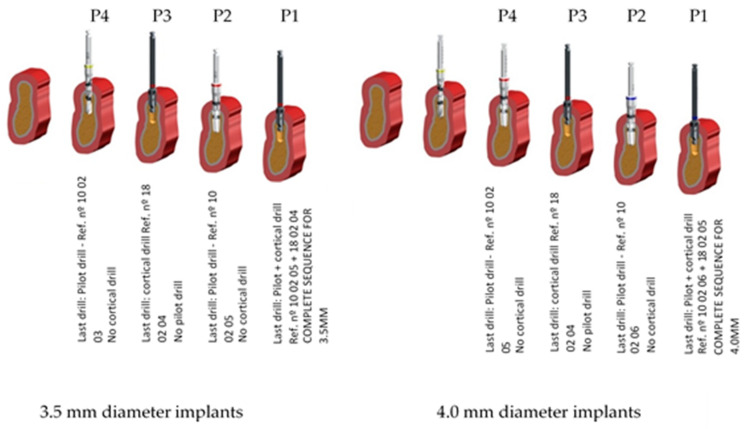
Three-dimensional morphology for P1, P2, P3 and P4 sequences of preparation sites for 3.5- and 4.0-mm diameter implants.

**Figure 4 ijerph-17-04436-f004:**
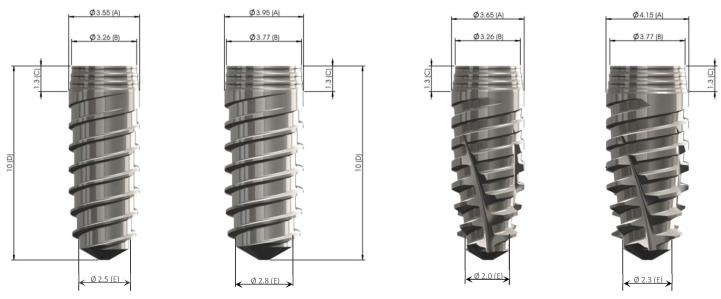
Macro-design of cylindrical (left) and tapered (right), both in 3.5- and 4.0-mm diameter.

**Table 1 ijerph-17-04436-t001:** Values of the Insertion Torque (IT, Newton/cm) and Implant Stability Quotient (ISQ) results for the different types of preparation technique for 3.5 mm and 4.0 mm diameter in both implant designs.

Prep.	Implant	IT—N/cm		ISQ	
		Mean	SD	Mean	SD
	3.5 mm				
P1	Cylindrical	34.6	15.4	74.9	5.3
	Tapered	54.2	22.6	78.2	7.5
P2	Cylindrical	44.1	12.4	78.2	6.5
	Tapered	57.3	15.2	78.6	3.9
P3	Cylindrical	63.7	18.7	76.4	6.8
	Tapered	75.9	16.2	79.0	7.3
P4	Cylindrical	66.4	24.4	76.2	11.2
	Tapered	68.8	21.7	79.6	5.2
	4.0 mm				
P1	Cylindrical	43.6	25.5	76.0	5.0
	Tapered	64.7	22.8	78.5	3.2
P2	Cylindrical	45.2	15.2	79.1	4.8
	Tapered	65.7	19.2	79.9	2.5
P3	Cylindrical	52.5	20.4	76.2	5.9
	Tapered	60.3	24.0	76.1	6.6
P4	Cylindrical	54.5	22.9	76.8	6.7
	Tapered	65.0	27.4	76.3	8.0

Prep: site preparations for each implant type and for each implant diameter; P1 to P4 see Section 2.6. *Study Groups*.

**Table 2 ijerph-17-04436-t002:** Statistical analysis comparing the Insertion Torque (IT) and Implant Stability Quotient (ISQ) for the two implant designs in the different types of preparation (Prep. Technique) for 3.5 mm and 4.0 mm (P1 to P4 see Section 2.6
*Study Groups*).

Implant Diameter	Prep. Technique	IT	Distribution		ISQ A	Distribution		ISQ B	Distribution		ISQ X
3.5 mm	P1	V < VX *p* = 0.000	V(N)	VX(NO)	V < VX *p* = 0.000	V(NO)	VX(NO)	V < VX *p* = 0.000	V(NO)	VX(NO)	V < VX
	P2	V < VX *p* = 0.000	V(N)	VX(NO)	V = VX *p* = 0.422	V(NO)	VX(NO)	V = VX *p* = 0.610	V(NO)	VX(NO)	V = VX
	P3	V < VX *p* = 0.002	V(N)	VX(NO)	V < VX *p* = 0.000	V(NO)	VX(NO)	V < VX *p* = 0.000	V(NO)	VX(NO)	V < VX
	P4	V = VX *p* = 0.711	V(NO)	VX(NO)	V = VX *p* = 0.082	V(NO)	VX(NO)	V = VX *p* = 0.351	V(NO)	VX(NO)	V = VX
4.0 mm	P1	V < VX *p* = 0.000	V(NO)	VX(NO)	V < VX *p* = 0.016	V(N)	VX(NO)	V < VX *p* = 0.039	V(N)	VX(NO)	V < VX
	P2	V < VX *p* = 0.000	V(NO)	VX(N)	V = VX *p* = 0.965	V(NO)	VX(NO)	V = VX *p* = 0.684	V(NO)	VX(NO)	V = VX
	P3	V = VX *p* = 0.127	V(N)	VX(N)	V = VX *p* = 1.000	V(NO)	VX(NO)	V = VX *p* = 0.348	V(NO)	VX(NO)	V = VX
	P4	V < VX *p* = 0.038	V(N)	VX(NO)	V = VX *p* = 0.358	V(NO)	VX(NO)	V = VX *p* = 0.942	V(NO)	VX(NO)	V = VX

V—Cylindrical; VX—Tapered X; V = VX—there are not statistical differences; V < VX—Tapered presents a higher value in the evaluated stability parameter; V > VX—Cylindrical presents a higher value in the evaluated stability parameter; V(N)—normal distribution; V(NO)—non normal distribution.

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
