# Peer review of "Influence of Implant Design and Under-Preparation of the Implant Site on Implant Primary Stability. An In Vitro Study"

_ijerph, 2020, doi:10.3390/ijerph17124436_

Round 1

Reviewer 1 Report

The paper is well done while the theme is well known.

All the study in conducted on one implant and does not give many informations regarding the other implant shapes.

I suggest to insert more informations regarding other similar studies conducted on other implants types and better motivate your choice...

Author Response

Response to the 1st Referee

Dear Referee,

Thank you very much for your comments, as they were extremely beneficial to the further development of this article. In that sense, we considered all your observations and suggestions, and made the necessary changes in the article. In the following lines, we will present you with the adjustments that were made.

“All the study in conducted on one implant and does not give many informations regarding the other implant shapes.

I suggest to insert more informations regarding other similar studies conducted on other implants types and better motivate your choice... “

A brief explanation of these types of studies and the citations have been included in the following lines of discussion (273-279)

We appreciate your attention,

Best regards.

Reviewer 2 Report

  1. Line 73 statement must be closed by the relevant citation.
  2. In introduction section the novelty of the work should be clearly mentioned. Describe why it was necessary to compare the effect on IPS of two different implants under the same drilling sequence?
  3. Line 102, authors should describe the shot blasting procedure in detail so that other people can compare their study with this manuscript?
  4. Line 102 authors must describe the passivation procedure and why it was necessary to passivate?
  5. Line 103, Ra should be first introduced as average roughness before using the abbreviation. Moreover, authors should describe how the roughness was measured which device used, what were the testing condition?
  6. Does the tapered design proved to be beneficial in the available literature also? What would be the reason for the better performance associated with the tapered design?
  7. Line 27-281 “The results found on this study are quite similar to the those found in Moon’s study, the under preparation group with the Tapered design implant had higher ISQ values” but what are the reasons for that.
  8. Again the reason for the particular behavior is missing, authors must clarify how drilling is beneficial in improving the stability of the implants (discussion needs to be sound).
  9. It is important to highlight that what are the areas for improvement and what are the clinical implication. How to use this study now into the clinical applications?

Author Response

Response to the 2nd Referee

Dear Referee,

Thank you very much for your comments, as they were extremely beneficial to the further development of this article. In that sense, we considered all your observations and suggestions, and made the necessary changes in the article. In the following lines, we will present you with the adjustments that were made.

  • Line 73 statement must be closed by the relevant citation.

The whole paragraph was removed

  • In introduction section the novelty of the work should be clearly mentioned. Describe why it was necessary to compare the effect on IPS of two different implants under the same drilling sequence?

The introduction was revised and it was described in the two last paragraphs this necessity (line 74-83)

  • Line 102, authors should describe the shot blasting procedure in detail so that other people can compare their study with this manuscript?

Line 102 authors must describe the passivation procedure and why it was necessary to passivate?

Line 103, Ra should be first introduced as average roughness before using the abbreviation. Moreover, authors should describe how the roughness was measured which device used, what were the testing condition?

Shotblasting and acid passivation procedures were described and explained the necessity of both protocols. SEM photography and the explanation of Ra meaning was also introduced.(line 96-105)

  • Does the tapered design proved to be beneficial in the available literature also? What would be the reason for the better performance associated with the tapered design?

It was introduced a brief explanation in the introduction of why the necessity of using a tapered implant and the benefits of this type of design. (line 67-73)

  • Line 273-281 “The results found on this study are quite similar to the those found in Moon’s study, the under preparation group with the Tapered design implant had higher ISQ values” but what are the reasons for that.

The reason for this to happen was explained in line 291-294

  • Again the reason for the particular behavior is missing, authors must clarify how drilling is beneficial in improving the stability of the implants (discussion needs to be sound).

It was introduced a paragraph explaining why the increase of IPS when there were a reduction of the preparation technique. (line 334-337)

  • It is important to highlight that what are the areas for improvement and what are the clinical implication. How to use this study now into the clinical applications?

A new topic named Clinical Implications explaning the clinical aplications was introduced (line 359-366)

We appreciate your attention,

Best regards.

Reviewer 3 Report

Dear Authors,

In general, the article is well written and has a reference to interesting research. The biomechanics of implant testing and the statistical module are clear and understandable. However, I have two reservations:

1) there is no reference to materials used for implants, especially comparison of mechanical properties, among others zirconia and titanium alloys made implants, which in my opinion is a serious lack because mechanics cannot be considered even in modeling in isolation from material properties, it would also be worth mentioning diseases such as bruxism that effectively impede implantation and negatively affect implant survival,

2) in the article it would seem to show at least a few photos from sample preparation, something from drilling or placing an implant, and it would be quite good to show a cross-sectional bone-implant connection, for example.

I recommend the article for acceptance after making the following changes:

1) please provide some detailed photos from sample preparation, it would be advisable to at least do a metallographic examination and show photos from a stereoscopic microscope even though the bone-implant connection alone, taking photos with SEM would be even better and definitely affect the reader's curiosity

2) please in the introduction refer to materials used for implants, show at least a comparison of ceramics and alloys of at least Ti6Al4V or new alloys with Nb and Zr, and refer to the problems of using implants - inflammation, loosening, bruxism, bone stiffening ou can quote The following articles that contain information on this topic:

1) Correlations between Sleep Bruxism and Temporomandibular Disorders, J. Clin. Med. 2020, 9 (2), 611; https://doi.org/10.3390/jcm9020611

2) Factors associated with abutment screw loosening in single implant supported crowns: A cross-sectional study https://doi.org/10.1016/j.mjafi.2018.06.011

3) Aging of Zirconia Dedicated to Dental Prostheses for Bruxers Part 1: Influence of Accelerating Aging for Surface Topography and Mechanical Properties https://doi.org/10.1515/rams-2019-0026

4) Aging of Zirconia Dedicated to Dental Prostheses for Bruxers Part 2: Influence of Heat Treatment for Surface Morphology, Phase Composition and Mechanical Properties https://doi.org/10.1515/rams-2019-0027

5) TEN-YEAR follow-up of treatment with zygomatic implants and replacement of hybrid dental prosthesis by ceramic teeth: A case report ttps: //doi.org/10.1016/j.amsu.2019.11.022

After making these changes - and especially improving the presentation of sample preparation, I recommend the article for acceptance.

Reviewer

Author Response

Response to the 3rd Referee

Dear Referee,

Thank you very much for your comments, as they were extremely beneficial to the further development of this article. In that sense, we considered all your observations and suggestions, and made the necessary changes in the article. In the following lines, we will present you with the adjustments that were made.

  • there is no reference to materials used for implants, especially comparison of mechanical properties, among others zirconia and titanium alloys made implants, which in my opinion is a serious lack because mechanics cannot be considered even in modeling in isolation from material properties, it would also be worth mentioning diseases such as bruxism that effectively impede implantation and negatively affect implant survival,

2nd Paragraph of introduction was introduced to refer some possible associated problems with implant therapy. (line 50-52)

  • in the article it would seem to show at least a few photos from sample preparation, something from drilling or placing an implant, and it would be quite good to show a cross-sectional bone-implant connection, for example.

A section with photos of the procedure was introduced in materials and methods – figure 2. (line 138)

I recommend the article for acceptance after making the following changes:

  • please provide some detailed photos from sample preparation, it would be advisable to at least do a metallographic examination and show photos from a stereoscopic microscope even though the bone-implant connection alone, taking photos with SEM would be even better and definitely affect the reader's curiosity

A section with photos of the procedure was introduced in materials and methods – figure 2. (line 138).

SEM photos were introduced with a brief explanation of the shotblasting and acid passivation procedures figure 1. (line 105).

2) please in the introduction refer to materials used for implants, show at least a comparison of ceramics and alloys of at least Ti6Al4V or new alloys with Nb and Zr, and refer to the problems of using implants - inflammation, loosening, bruxism, bone stiffening ou can quote The following articles that contain information on this topic:

2nd Paragraph of introduction was introduced to refer some possible associated problems with implant therapy, with supporting references. (line 50-52)

We appreciate your attention,

Best regards.

Reviewer 4 Report

Dear authors,

Congratulations on your study, it is very interesting and I believe it deserves to be published.
However it must be improved in its presentation and analysis.
The entire manuscript requires an extensive proof-read for English.
The first paragraph of materiai and methods is a repetition of the last of the introduction, it must be deleted.

The introduction of two different diameters is a variable that, just like the preparation technique and the shape of the implant (tapered or not), can influence the result. However, it seems to me that it was not taken into consideration as a variable in the analysis.

Statistical analysis is confusing. What is reported in materials and methods is perhaps a useful summary for a small statistical manual, however it is not described which statistical method is actually used in the article. It is clear that according to the normal or not distribution of the data, a parametric or non-parametric method will be used, but which one was actually used in the end?

The results do not specify which statistical method was used, whether an anova or a kruskal-wallis. However, since multiple dependent variables interact with multiple independent variables, it would probably have been more useful to use a two-way ANOVA. In any case, to understand the significance of the differences between the individual groups, a post-hoc test (Tukey, or whatever you prefer) would also be necessary. I suggest once again to insert the diameter as an additional variable.
You have all the data, it will not be difficult for you to perform the statistical analysis again, but I suggest you use the advice of an expert in statistics.

Thanks again for submitting this article and good luck.

Best regards,

Author Response

Response to the 4th Referee

Dear Referee,

Thank you very much for your comments, as they were extremely beneficial to the further development of this article. In that sense, we considered all your observations and suggestions, and made the necessary changes in the article. In the following lines, we will present you with the adjustments that were made.

  • However it must be improved in its presentation and analysis.
    The entire manuscript requires an extensive proof-read for English.
    The first paragraph of materiai and methods is a repetition of the last of the introduction, it must be deleted.

The last paragraph of introduction was deleted.

  • The introduction of two different diameters is a variable that, just like the preparation technique and the shape of the implant (tapered or not), can influence the result. However, it seems to me that it was not taken into consideration as a variable in the analysis.

  1. Statistical analysis is confusing. What is reported in materials and methods is perhaps a useful summary for a small statistical manual, however it is not described which statistical method is actually used in the article. It is clear that according to the normal or not distribution of the data, a parametric or non-parametric method will be used, but which one was actually used in the end?

It was included in table 2. The type of the data distribution (line 253)

The results do not specify which statistical method was used, whether an anova or a kruskal-wallis. However, since multiple dependent variables interact with multiple independent variables, it would probably have been more useful to use a two-way ANOVA. In any case, to understand the significance of the differences between the individual groups, a post-hoc test (Tukey, or whatever you prefer) would also be necessary. I suggest once again to insert the diameter as an additional variable.
You have all the data, it will not be difficult for you to perform the statistical analysis again, but I suggest you use the advice of an expert in statistics.

The ANOVA method is part of the parametric statistical methods, which require the fulfillment of a series of assumptions, including the normal distribution of the results. This assumption was not fulfilled in some of the study data. In the event that any of the distributions does not fit a normal distribution, non-parametric statistical methods must be used, so if two distributions are being compared, the Mann Whitney method must be used, and if more are being compared of two distributions, the Kruskall Wallis method should be used.

Christian Heumann, Michael Schomaker, Shalabh, Introduction to Statistics and Data Analysis, Springer 2016 ISBN: 978-3-319-46160-1

Massimiliano Bonamente, Statistics and Analysis of Scientific Data, Springer 2017, ISBN: 978-1-4939-6570-0

Bewick V, Cheek L, Ball J. Statistics review 10: further nonparametric methods. Crit Care. 2004;8(3):196-9.

Whitley E, Ball J. Statistics review 6: Nonparametric methods. Crit Care. 2002;6(6):509-1

We appreciate your attention,

Best regards.

Round 2

Reviewer 2 Report

The manuscript has been revised appropriately.

Author Response

thank you very much for your help. It has been a pleasure working under your instructions.

Reviewer 3 Report

Thank you for following the suggestion. I recommend the article for acceptance.

Best Regards

Author Response

(The authors gave the same response as above.)

Reviewer 4 Report

Dear authors,

I suggested to delete the first paragraph of materials and method, not the last of introduction, where it's normally good to declare the aims.

Thanks for the little explanation of what ANOVA and Mann-Whitney is, however, I have some experience with statistics...

Question about diameter was disregarded.

Best regards,

Author Response

Dear Referee,

We would like to thank you again for your comments, and apologize for the misinterpretation and communication problem between the authors. In the following lines, we will present the last changes made according to your observations.

  • I suggested to delete the first paragraph of materials and method, not the last of introduction, where it's normally good to declare the aims.

We regret that in the first review, the repetition of the paragraph was eliminated but not in the most appropriate way. We correct it following your instructions. (line 84-87)

  • Statistical analysis is confusing. What is reported in materials and methods is perhaps a useful summary for a small statistical manual, however it is not described which statistical method is actually used in the article. It is clear that according to the normal or not distribution of the data, a parametric or non-parametric method will be used, but which one was actually used in the end?

We did not really understand at first the scope of the suggestion. We regret our error and acknowledge that the method used was not actually specified. The statistical material and method are modified with a new wording. (line 203-214) We appreciate your effort to improve the study.

  • The introduction of two different diameters is a variable that, just like the preparation technique and the shape of the implant (tapered or not), can influence the result. However, it seems to me that it was not taken into consideration as a variable in the analysis.

Actually, in Table 2 and in its legend, the statistical meanings of the different diameters are specified. In any case, the working hypothesis is that stability can be improved as a function of undercutting, and that this improves primary stability (in our case in both diameters, 3.5 and 4.0). In future investigations we will be able to evaluate the importance of the diameter and other factors of the implant such as micro and macroscopic morphology, but initially it was not the original proposal. We take note of your suggestion for our line of work and we are very grateful to you.

It was fortunate to be able to count on your help. Clearly the article has improved.

Thank you again.